# Empowering adult patients with diabetes for health educators' role within their family members: A cross-sectional study

**Mona Alanazi**[1,2,3]*, **Eman Bajmal**[1,2,3], **Abeer Aseeri**[4], **Ghaida Alsulami**[5]

1 College of Nursing, King Saud bin Abdulaziz University for Health Sciences, Riyadh, Saudi Arabia, 2 King Abdullah International Medical Research Center, Riyadh, Saudi Arabia, 3 Ministry of the National Guard Health Affairs Riyadh, Riyadh, Saudi Arabia, 4 Nursing College-Muhyil Asir, King Khalid University, Abha, Saudi Arabia, 5 Nursing College, Umm Al-Qura University, Mecca, Saudi Arabia

* enizimo@ksau-hs.edu.sa

## Abstract

**Data Availability Statement:** The datasets used and/or analyzed during the current study are available as attachment in the Supporting Information files.

### Background

Patient engagement as partners in diabetes prevention for family members/close relatives is a novel and underexplored approach. This paper aims to assess patients' willingness and confidence in their ability to succeed as health educators for their family members and investigate the influencing factors.

### Methods

A cross-sectional descriptive study was conducted between January 2023 and April 2023. A newly developed and validated self-reported questionnaire, based on the Health Belief Model (HBM) and previous research, was administered to a convenient sample of 134 adult ~~participants~~ diagnosed with diabetes. These participants sought care at primary healthcare clinics at King Abdul-Aziz Medical City, Ministry of National Guard Health Affairs in Riyadh and Jeddah (MNGHA). The data was examined using statistical methods including descriptive analysis, ANOVA, Tukey's HSD (Honestly Significant Difference) Post Hoc tests, and Pearson's correlation coefficients.

### Results

The majority of participants expressed a willingness to assume the role of health educators for their family members (n = 117, 87.31%) and reported a high level of willingness and confidence, as indicated by self-efficacy scores ranging from 12.00 to 25.00, with a mean of 21.12 (SD = 2.76). Participants' willingness to be health educators exhibited positive correlations with their perceptions of diabetes severity and susceptibility (r = .433, p < .01), perceived benefits and barriers (r = .451, p < .01), cues to action (r = .520, p < .01), self-efficacy (r = .789, p < .01), and the total score of the questionnaire (r = .640, p < .01).

**Funding:** The author(s) received no specific funding for this work.

**Competing interests:** The authors have declared that no competing interests exist.

## Conclusions

The majority of participants expressed their willingness to assume the role of health educators for their family members, and a significant portion reported confidence in their capacity to accomplish this objective. Healthcare providers should emphasize the importance of equipping patients with the skills and knowledge necessary to effectively convey health messages and serve as health educators within their communities. This expansion of the approach holds the potential to have a significant impact on public health strategies for diabetes prevention.

## Introduction

In the realm of healthcare, addressing the growing concerns surrounding diabetes has become increasingly important [1]. Among the innovative strategies emerging to tackle this global health issue, a particularly novel approach involves harnessing the power of those who are already familiar with the challenges of diabetes–patients themselves [2,3]. This innovative concept entails enlisting individuals living with diabetes to serve as health educators within their local communities [2]. By leveraging their experiences, these patients can offer unique insights, empathy, and practical advice to others who may be at risk of developing diabetes or struggling to manage the condition [4]. This approach not only enhances the understanding of diabetes within communities but also fosters a sense of empowerment, solidarity, and shared responsibility in the battle against this chronic illness [3,5].

However, the key question revolves around the feasibility of employing the "patients as health educators" approach to target high-risk individuals in Saudi Arabia. This approach becomes especially pertinent in light of the high prevalence of familial aggregation of diabetes within Saudi families, making it seemingly appropriate to leverage patients as health educators to educate their family members as a preventive strategy aimed at high-risk individuals [3,5,6]. In the context of Saudi Arabia, addressing diabetes takes on added significance due to its high prevalence within families. Engaging patients as educators could directly impact these familial clusters at higher risk. The need to explore the feasibility of this approach lies in its potential to empower and educate within these specific family units, potentially curbing the concerning rise of diabetes in our community. This aligns global strategies with targeted local interventions, emphasizing the potential impact within our specific familial context.

Nonetheless, several critical factors must be taken into account before implementing this approach. Patients' beliefs, cultural backgrounds, and sociodemographic characteristics may exert a significant influence on various aspects related to diabetes prevention. These include their level of knowledge about diabetes, their perception of family risk, the seriousness attributed to diabetes, concerns regarding family health, and their willingness to engage in health education efforts [7,8]. These factors raise legitimate concerns about the effectiveness of patients serving as health educators to their family members and relatives in the context of diabetes prevention.

### Theoretical framework

The Health Belief Model (HBM) provides a theoretical foundation for exploring the factors that may affect individuals' decisions to act as health educators within their families. These factors include their beliefs about diabetes, perceived benefits and barriers to health education,

self-efficacy in performing health education tasks, and concerns about their family members' susceptibility to diabetes [8]. The presence of cues to action, as proposed by the HBM, can also play a role in motivating individuals to engage in health-promoting behaviors, including health education for their family members [9,10].

The proposed theoretical framework, based on the HBM and existing literature on diabetes prevention, aims to understand and explain why individuals with diabetes may choose to educate their family members about diabetes prevention. It suggests that individuals are more likely to take on this role if they perceive diabetes as a serious health problem, believe that health education can reduce the risk and severity of diabetes among their family members, have confidence in their ability to effectively educate, and are concerned about their family members' health [8,10]. Fig 1 explains the substruction of the theoretical framework of HBM (Fig 1).

Therefore, the aims of the study were to assess the patient's willingness and confidence in their ability to succeed as health educators for their family members and investigate the influencing factors.

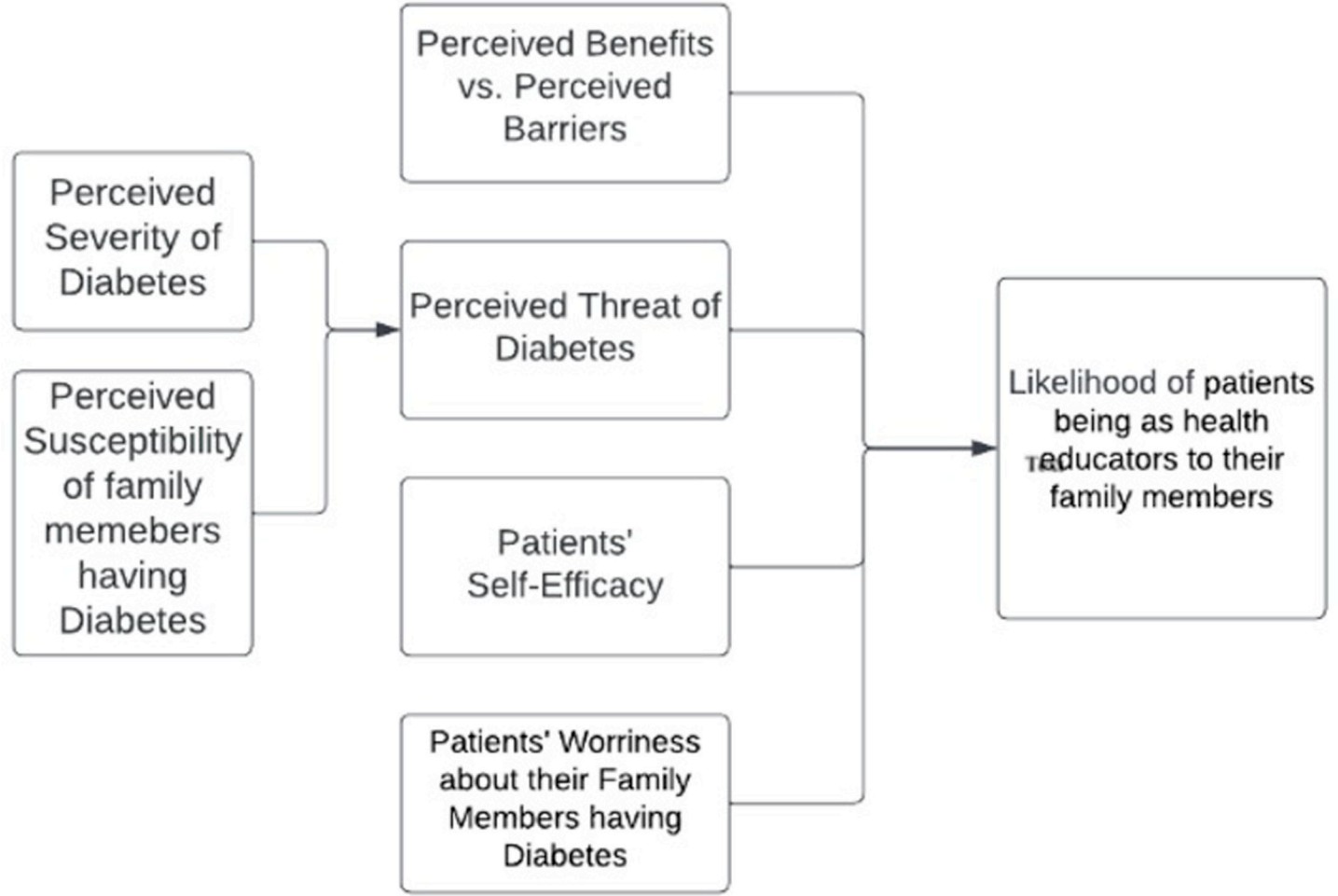

**Fig 1. Pictorial representation of the substruction model.**

## Materials and methods

### Study design

To achieve the study objectives, a cross-sectional descriptive study design was employed. The preparation of this report adhered to the Strengthening the Reporting of Observational Studies in Epidemiology (STROBE) checklist for cross-sectional studies [11].

### Setting and participants

This research was conducted from January 2023 to April 2023. A convenience sample was recruited from the Primary Health Clinics (PHC) at King Abdul-Aziz Medical City, Ministry of National Guard Health Affairs in Riyadh and Jeddah cities (MNGHA). Inclusion criteria consisted of Saudi adult patients diagnosed with any type of diabetes who were Arabic speakers. Adults who were unable to read the questionnaire were not included. All eligible participants were invited to join the study voluntarily. The sample size was calculated using G*Power software, which is designed for precise sample size analysis and power calculations. The goal was to enroll n = 122 participants to achieve a statistical power of 95%. This calculation was based on a medium effect size of 0.3, an alpha error probability of 5%, and an estimated 10% rate of missing data. Consequently, the overall sample size was determined as N = (122 participants + 12 accounting for anticipated missing data), resulting in a total of 134 participants.

### Recruitment

A recruitment flyer with the questionnaire barcode was distributed to all patients at the chosen site by the research team. Those who were interested in participating were guided either to access the questionnaire link online and complete the survey or to complete a paper-based questionnaire.

### Instrument

Data were collected for this study using the newly developed questionnaire based on the HBM and previous studies. The questionnaire included two parts.

**Part one: Socio-demographics and diabetes-related characteristics.** Socio-demographics include age, self-reported gender, marital status, offspring, number of offspring, family type, role within the family, monthly income, and educational level. Diabetes-related characteristics include a family history of diabetes in first- and/or second-degree relatives, diabetes duration (less or more than ten years), treatment (diet, tablets, insulin), diabetes complications (diabetes causing problems with eyes, feet, and/or kidneys), comorbidities, including cardiovascular problems, previous diabetes-related education.

**Part two: Patients' willingness, confidence levels, and influencing factors.** Based on the HBM's factors and previous studies, participants were asked about different dimensions to assess their willingness and confidence in their ability to succeed as health educators for their family members and to investigate the influencing factors [2–5,7,10]. Responses are based on their agreement on a 5-point Likert scale ranging from "Strongly agree" to "Strongly disagree". The total score varies from 14 to 70; a higher score indicates a smooth application of this approach, and a lower score indicates the need for interventions to prepare the participants to act as health educators. The cut of point is 45, which indicates the need for interventions.

*Self-efficacy*. Participants' willingness and confidence in their ability to succeed as health educators for their family members (self-efficacy) were assessed by asking the participants to rate their agreement on the following items: (1) They already initiated diabetes prevention discussions with their family members and close relatives, (2) they have the ability to act as a

health educator, (3) health education preparation will increase their confidence as health educators, (4) they must receive professional education before they act as health educators, and (5) they are willing to be a health educator for their family member to prevent diabetes. The willingness of patients to serve as health educators was also evaluated through a separate binary question. Participants were asked to respond with a "yes" or "no" to indicate whether they were open to acting as health educators for their families and close relatives.

*Perceived severity and susceptibility*. To assess the participants' belief of the seriousness of diabetes and the perceived threat of diabetes, they were asked to rate their agreement on the severity of diabetes. Moreover, they were asked to rate the likelihood that their family members and close relatives will develop diabetes.

*Perceived benefits and perceived barriers*. To assess the possibility of preventing or delaying diabetes onset among family members (perceived benefits), participants were asked to rate their agreement on the following items if they were engaged in educating their family members: (1) make their family members more aware of the importance of lifestyle changes to prevent diabetes, (2) encourage family members to make lifestyle changes, and (3) help their family members in preventing diabetes. With regard to perceived barriers to action, participants were asked to rate their agreement on the following items: (1) they are knowledgeable regarding diabetes prevention education, (2) they know how to inform their family members about diabetes prevention, and (3) they know which family members to educate.

*Cues to action*. To measure the exposure to factors that prompt action (cues to action), participants were asked to rate their worries about their family members developing diabetes.

## Reliability and validity

The first stage of this study involved piloting questionnaires to assess their validity, reliability, and feasibility. Pilot testing was conducted among 10% of the total required sample size and they were not included in the analysis of this paper. The aim of the pilot testing was to evaluate the questionnaire's performance, identify potential issues, and gather initial data for assessing its reliability and validity.

The validity and reliability of the questionnaire were assessed using established statistical methods, including Cronbach's alpha, internal consistency, and content validity.

Cronbach's alpha is a commonly used measure of internal consistency reliability [12]. It assesses how well the items in a questionnaire or scale are correlated with each other. As shown in the results section of this research, the questionnaire revealed a good Cronbach's alpha value which indicates good internal consistency, implying that the items in the questionnaire measure the same construct, referring to Table 3.

Content validity is indeed a crucial aspect of questionnaire development, ensuring that the items effectively measure the intended construct [13,14]. In the current study, content validity was assessed through a process involving four experts and researchers who were knowledgeable about the subject matter. Therefore, four experts and researchers who were familiar with the topic of the questionnaire were selected to evaluate the content of the questionnaire. These individuals are considered experts in the field, making them well-qualified to assess the relevance and comprehensiveness of the questionnaire items. The experts and researchers reviewed the questionnaire items to determine whether they were relevant to the construct being measured. They assessed whether the questions accurately captured the key aspects of the subject matter under investigation. In addition to assessing relevance, the experts also evaluated the questionnaire's comprehensiveness. They ensured that the items covered a broad range of aspects related to the subject matter, leaving no critical components unaddressed. To facilitate the evaluation process, a 4-point ordinal rating relevance scale was utilized [15]. This

scale likely allowed the experts to provide structured feedback on the relevance of each item, helping to quantify their judgments. Based on the feedback and ratings provided by the experts, any items that were deemed irrelevant or insufficiently comprehensive could be adjusted or revised to enhance content validity, refer to Table 3.

## Statistical analysis

Upon completion of data collection, researchers input the data into an Excel database for subsequent analysis. Statistical Package for the Social Sciences computer software (SPSS for Mac, Version 21.0) was utilized for data analysis.

Regarding patients' willingness to become health educators, responses were reassessed and categorized into 'Yes' or 'No'. 'Strongly agree' and 'Agree' responses were grouped as 'Yes', while 'Neutral', 'Disagree', or 'Strongly disagree' were classified as 'No'.

In this study, different analyses were employed considering patients' willingness to become health educators as the outcome variable. Descriptive statistics—means, standard deviations, frequencies, and percentages—were computed for all variables of interest to characterize the study variables. One-way analysis of variance (ANOVA) and Tukey's HSD (Honestly Significant Difference) Post Hoc tests were applied to examine mean differences between demographic characteristics (marital status, family type, participant's role in the family, employment status, education level, and family income) and study variables (perceived severity and susceptibility, benefits, barriers, self-efficacy, cues to action and scale total mean score). Additionally, Pearson's correlation coefficients were utilized to investigate relationships among continuous variables. The threshold for statistical significance was set at the standard alpha level of .05.

## Ethics statement

Institutional Review Board (IRB) approval was obtained from King Abdullah International Medical Research Center (KAIMRC) with IRB No NRC22R/336/08. Various measures were undertaken to protect study participants' rights and assure confidentiality. Participants' written informed consent to participate in the study was presumed, and no personally identifiable information was collected.

## Results

### Demographic characteristics of the participants

A total of 134 participants were recruited from various primary healthcare centers affiliated with the Ministry of National Guard Health Affairs (NGHA). The participants' ages ranged from 25 to 80 years, with a mean of 52.32 ($SD$ = 13.23) years. The majority of participants were female (n = 78, 57.8%), while males constituted a smaller proportion (n = 56, 41.5%). A significant percentage of participants, 88.9%, reported having children, whereas only 10.4% did not. Regarding family structure, the majority of participants (n = 69, 51.1%) resided in nuclear families, whereas the fewest participants lived in blended families (n = 5, 3.70%). Approximately 82.2% of participants received health education about diabetes mellitus (DM) from healthcare professionals, including physicians, nurses, and dieticians. For more detailed information about the sample characteristics, please refer to Table 1.

### Patients' willingness and confidence level of health educators

The majority of participants expressed their willingness to serve as health educators for diabetes prevention among their family members and close relatives (n = 117, 87.31%), as shown in Table 1. In terms of self-efficacy scores, they reported from 12.00 to 25.00, with a mean of

**Table 1. Demographic characteristics of the participants and association between characteristics and scale items (N = 134).**

| Demographic variable | | | Total score | Perceived susceptibility/ severity | Perceived benefits/barriers | Self-efficacy | Cues to action |
|---|---|---|---|---|---|---|---|
| | n | % | M ± SD | M ± SD | M ± SD | M ± SD | M ± SD |
| Age | | | | | | | |
| <30 | 7 | 5.22 | 61.14±9.15 | 8.42±1.98 | 26.42±4.35 | 21.85±2.54 | 4.42±.78 |
| 30–40 | 23 | 17.16 | 60.04±7.99 | 8.17±1.46 | 25.65±3.62 | 21.65±2.56 | 4.56±.66 |
| 40–50 | 34 | 25.37 | 56.70±4.53 | 7.94±.95 | 23.50±2.68 | 20.79±1.78 | 4.47±.74 |
| 50–60 | 37 | 27.61 | 60.27±6.79 | 8.36±1.53 | 25.22±3.26 | 21.97±2.68 | 4.72±.51 |
| 60–70 | 23 | 17.16 | 56.00±9.98 | 8.21±1.56 | 23.82±4.39 | 19.73±3.97 | 4.21±.67 |
| >70 | 10 | 7.46 | 57.81±5.65 | 8.27±.90 | 25.00±2.19 | 20.72±2.05 | 3.81±.75 |
| p-value | | | .115 | .861 | .076 | .045 | .003* |
| Gender | | | | | | | |
| Male | 56 | 41.5 | 58.89±6.76 | 8.35±1.36 | 24.94±3.13 | 21.19±2.56 | 4.39±.65 |
| Female | 78 | 57.8 | 58.11±7.77 | 8.07±1.36 | 24.46±3.70 | 21.07±2.91 | 4.50±.73 |
| p-value | | | .311 | .692 | .247 | .542 | .626 |
| Marital Status | | | | | | | |
| Single | 10 | 7.40 | 63.10±4.50 | 8.80±.632 | 26.90±2.80 | 22.50±1.43 | 4.90±.31 |
| Married | 108 | 80.0 | 58.37±7.25 | 8.18±1.43 | 24.65±3.46 | 21.10±2.61 | 4.43±.68 |
| Widowed | 8 | 5.90 | 52.62±10.23 | 7.00±.755 | 21.62±3.73 | 19.87±5.56 | 4.12±1.12 |
| Divorced | 8 | 5.90 | 59.25±4.62 | 8.75±.707 | 25.00±2.07 | 21.00±1.69 | 4.50±.53 |
| p-value | | | .026 | .023 | .007* | .250 | .114 |
| Having Children | | | | | | | |
| Yes | 120 | 88.9 | 58.09±7.33 | 8.14±1.39 | 24.49±3.43 | 21.03±2.78 | 4.42±.71 |
| No | 14 | 10.4 | 61.42±7.10 | 8.64±1.00 | 26.14±3.54 | 21.92±2.52 | 4.71±.46 |
| p-value | | | .751 | .116 | .794 | .506 | .027 |
| Family Type | | | | | | | |
| Nuclear Family | 69 | 51.1 | 57.63±7.00 | 7.97±1.41 | 24.21±3.49 | 20.94±2.44 | 4.50±.67 |
| Single Family | 11 | 8.10 | 57.45±11.59 | 8.27±1.34 | 24.81±5.32 | 20.00±4.42 | 4.36±1.02 |
| Extended Family | 49 | 36.3 | 59.65±6.83 | 8.42±1.32 | 25.26±2.94 | 21.57±2.75 | 4.38±.67 |
| Blended Family | 5 | 3.70 | 60.25±6.55 | 9.00±.81 | 22.25±3.40 | 21.50±2.64 | 4.50±.57 |
| p-value | | | .633 | .320 | .509 | .420 | .806 |
| Participant's role in the family | | | | | | | |
| Son | 9 | 6.70 | 59.22±6.49 | 8.00±.92 | 25.55±1.36 | 21.22±2.10 | 4.55±.52 |
| Daughter | 4 | 3.00 | 63.50±4.72 | 9.25±.50 | 26.75±3.35 | 22.50±1.91 | 5.00±.00 |
| Mother | 60 | 44.4 | 57.26±8.32 | 8.06±1.50 | 24.10±2.36 | 20.68±3.01 | 4.41±.78 |
| Father | 53 | 39.3 | 58.69±6.70 | 8.33±1.30 | 24.92±3.93 | 21.07±2.54 | 4.35±.65 |
| Grandmother | 6 | 4.40 | 60.83±1.60 | 7.33±.51 | 24.50±3.15 | 24.00±1.67 | 5.00±.00 |
| Grandfather | 2 | 1.50 | 66.00±.00 | 10.00±.00 | 27.00±1.37 | 24.00±.000 | 5.00±.00 |
| p-value | | | .276 | .007* | .442 | .045 | .122 |
| Employment Status | | | | | | | |
| Full-time | 53 | 39.3 | 59.43±7.13 | 8.50±1.26 | 25.24±3.33 | 21.20±2.66 | 4.47±.66 |
| Part-time | 13 | 9.60 | 55.07±10.91 | 7.92±1.49 | 23.07±4.68 | 19.61±4.55 | 4.46±.66 |
| Unemployed | 29 | 21.5 | 56.48±6.83 | 7.34±1.49 | 23.86±3.18 | 20.82±2.12 | 4.44±.90 |
| Retired | 37 | 27.4 | 59.27±6.01 | 8.43±1.09 | 24.81±3.23 | 21.62±2.44 | 4.40±.59 |
| Student | 2 | 1.50 | 67.00±4.24 | 9.50±.70 | 28.50±2.12 | 24.00±1.41 | 5.00±.00 |
| p-value | | | .060 | .001* | .081 | .105 | .843 |
| Education level | | | | | | | |
| Uneducated | 26 | 19.3 | 58.80±4.96 | 8.00±1.05 | 23.76±3.19 | 22.23±1.92 | 4.80±.63 |

*(Continued)*

**Table 1.** (Continued)

| Demographic variable | | | Total score | Perceived susceptibility/ severity | Perceived benefits/barriers | Self-efficacy | Cues to action |
|---|---|---|---|---|---|---|---|
| | n | % | M ± SD | M ± SD | M ± SD | M ± SD | M ± SD |
| High School | 27 | 20.0 | 56.62±7.01 | 7.44±1.62 | 23.88±3.20 | 20.85±2.21 | 4.44±.69 |
| High Diploma | 24 | 17.8 | 54.54±8.33 | 8.08±1.47 | 22.80±3.02 | 19.08±3.33 | 4.04±.69 |
| Bachelor's Degree | 35 | 25.9 | 60.42±6.80 | 8.57±1.03 | 25.82±3.17 | 21.65±2.55 | 4.37±.64 |
| Master's Degree | 20 | 14.8 | 60.45±7.88 | 8.75±1.29 | 25.90±3.52 | 21.20±2.78 | 4.60±.68 |
| Ph.D. Degree | 2 | 1.5 | 70.00±.000 | 10.00±.000 | 30.00±.000 | 25.00±.000 | 5.00±.00 |
| p-value | | | .003* | .002* | .002* | < .001* | .003* |
| Family income | | | | | | | |
| < SR5,000 | 19 | 14.1 | 59.05±4.70 | 8.00±.94 | 24.73±2.64 | 21.73±1.91 | 4.57±.76 |
| SR5,000-SR10,000 | 44 | 32.6 | 55.75±8.18 | 7.70±1.51 | 23.06±3.75 | 20.50±3.44 | 4.47±.69 |
| SR10,000-SR15,000 | 34 | 25.2 | 57.88±7.50 | 8.08±1.42 | 24.85±3.36 | 20.76±2.58 | 4.17±.75 |
| SR15,000-SR20,000 | 27 | 20.0 | 61.11±5.91 | 8.92±.95 | 25.88±2.77 | 21.66±2.07 | 4.62±.56 |
| > SR20,000 | 10 | 7.40 | 63.80±5.99 | 9.10±.99 | 27.60±2.59 | 22.50±2.50 | 4.60±.51 |
| p-value | | | .004* | < .001* | < .001* | .121 | .085 |
| Participant's willingness to be a health educator | | | | | | | |
| Yes | 117 | 87.31 | 60.09±5.27 | 8.42±1.12 | 25.31±2.81 | 21.79±1.63 | 4.55±.56 |
| No | 17 | 12.68 | 42.70±2.87 | 6.30±.81 | 18.3±1.50 | 14.30±.72 | 3.80±.36 |
| p-value | | | < .001* | < .001* | < .001* | < .001* | < .001* |

21.12 (SD = 2.76), signifying a high level of willingness and confidence among participants regarding their ability to succeed as health educators for their family members, as detailed in Table 3.

## The influencing factors of willingness of health educators

The influencing factors on the willingness of health educators were investigated in this study. Significant differences in education levels were observed concerning both the total scale score and self-efficacy, with F (5, 128) = 3.865, p = .003, and F (5, 128) = 5.269, p < .001, respectively. Participants holding high school diploma certificates exhibited notably lower total mean scores (M = 54.54, SD = 8.33) and lower self-efficacy mean scores (M = 19.08, SD = 3.33) compared to those with lower or other levels of education. Additionally, significant disparities in family income were identified in relation to the total scale score, with F (4, 129) = 4.128, p = .004. Participants with family incomes between SR5,000 and SR10,000 had significantly lower mean scores (M = 55.75, SD = 8.18) than participants with family incomes below SR5,000 or above SR10,000.

With regard to the perceived susceptibility/severity domain, there was a significant difference based on the participant's role within the family, employment status, education level, and family income. As shown in Table 1, there were significant differences for the participant's role within the family and perceived severity and susceptibility, F (5, 128) = 3.46, p = .007. Grandmother participants had significantly lower mean scores (M = 7.33, SD = .51) than being son (M = 8.00, SD = .92), daughter (M = 9.25, SD = .50), mother (M = 8.06, SD = 1.50), father (M = 8.33, SD = 1.30), and grandfather participants (M = 10.00, SD = .000). Moreover, there were significant employment status differences with perceived severity and susceptibility, *F* (4, 129) = 4.869, *p* = .001. Unemployed participants had significantly lower mean scores (M = 7.34, SD = 1.49) than full-time employee (M = 8.50, SD = 1.26), part-time employee (M = 7.92, SD = 1.49), retired (M = 8.43, SD = 1.09), and student participants (*M* = 9.50, SD =

.70). In addition, there were significant education level differences with perceived severity and susceptibility, $F$ (5, 128) = 4.068, $p$ = .002. Participants with high school certificate had significantly lower mean scores ($M$ = 7.44, $SD$ = 1.62) than uneducated/other level of education participants. Lastly, there were significant family income differences with perceived severity and susceptibility, F (4, 129) = 5.155, p < .001. Participants with family income between SR5,000-SR10,000 had significantly lower mean scores (M = 7.70, SD = 1.51) than participants whose family income less than SR5,000 and more than SR10,000.

With regard to the perceived benefits/barriers domain, there was a significant difference based on marital status, education level, and family income. As shown in Table 1, there were significant marital status differences with perceived benefits and barriers, F (3, 130) = 4.194, p = .007. Widowed participants had significantly lower mean scores (M = 21.62, SD = 3.73) than single (M = 26.90, SD = 2.80), married (M = 24.65, SD = 3.46) and divorced participants (M = 25.00, SD = 2.07). In addition, there were significant education level differences with perceived benefits and barriers, F (5, 128) = 3.947, p = .002. Participants with high diploma certificates had significantly lower mean scores (M = 22.80, SD = 3.02) than uneducated/other level of education participants. Also, there were significant family income differences with perceived benefits and barriers, F (4, 129) = 5.669, p < .001. Participants with family income between SR5,000-SR10,000 had significantly lower mean scores (M = 23.06, SD = 3.75) than participants whose family income less than SR5,000 and more than SR10,000.

With regard to cues to action domain, there was a significant difference based on the participants age and education level.

As shown in Table 1, there were significant participants age differences with cues to action, F (5, 128) = 3.899, p = .003. Participants older than 70 years had significantly lower mean scores (M = 3.81, SD = .75) than those participants younger than 70 years. In addition, there were significant educational level differences with cues to action, F (5, 128) = 3.886, p = .003. Participants with high diploma certificate had significantly lower mean scores (M = 4.04, SD = .69) than uneducated/other level of education participants.

In the correlation analysis presented in Table 2, participants' willingness to serve as health educators for their family members and close relatives demonstrated positive and moderate correlations with perceived severity and susceptibility (r = .433, p < .01), perceived benefits and barriers (r = .451, p < .01), and cues to action (r = .520, p < .01). Conversely, participants' willingness to take on this role exhibited a strong positive correlation with self-efficacy (r = .789, p < .01). Furthermore, self-efficacy was positively and moderately correlated with perceived severity and susceptibility (r = .678, p < .01) and cues to action (r = .633, p < .01), as well as positively and strongly correlated with perceived benefits and barriers (r = .720, p < .01).

**Table 2. Pearson's product–moment correlations.**

| Variable | 1 | 2 | 3 | 4 | 5 | 6 | 7 |
|---|---|---|---|---|---|---|---|
| 1-Age | — | | | | | | |
| 2-Total score | -.092 | — | | | | | |
| 3-Perceived Severity and Susceptibility. | .027 | .858** | — | | | | |
| 4-Perceived Benefits and Perceived Barriers. | -.071 | .932** | .786** | — | | | |
| 5-Cues to Action. | -.202* | .631** | .480** | .440** | — | | |
| 6-Self-Efficacy. | -.117 | .903** | .678** | .720** | .633** | — | |
| 7-Participant's willingness to be a health educator | -.166 | .640** | .433** | .451** | .520** | .789** | — |

Note: *p < .05

**p < .01.

The questionnaire scores, as outlined in Table 3, showcased the following statistics: total scores ranged from 38 to 70, with a mean of 58.44 (SD = 7.35); perceived severity and susceptibility scores ranged from 4.00 to 10.00, with a mean of 8.19 (SD = 1.36); perceived benefits and barriers scores ranged from 16.00 to 30.00, with a mean of 24.66 (SD = 3.47); self-efficacy scores ranged from 12.00 to 25.00, with a mean of 21.12 (SD = 2.76); and cues to action scores ranged from 2.00 to 5.00, with a mean of 4.45 (SD = 0.70).

## Reliability and validity

In Table 3, the assessment of internal consistency using Cronbach's alpha coefficient revealed satisfactory reliability measures for the scales employed in this study. The 2-item perceived severity and susceptibility scale achieved a Cronbach's alpha coefficient of 0.79, with item total correlation coefficients spanning from 0.14 to 0.85. Similarly, the 6-item perceived benefits and barriers scale demonstrated a Cronbach's alpha coefficient of 0.79, accompanied by item-total correlation coefficients ranging from 0.076 to 0.81. The 5-item self-efficacy scale achieved a Cronbach's alpha coefficient of 0.77, with item total correlation coefficients ranging from 0.10 to 0.78, which meets acceptable reliability standards. Additionally, the 1-item cues to the action scale exhibited a Cronbach's alpha coefficient of 1.00.

For content validity assessment, a 4-point ordinal rating relevance scale was employed to evaluate the appropriateness of the scale's items (Lenz, 2020). As detailed in Table 3, the inter-rater reliability for both the total scale score and the four individual domains exceeded 75%. Furthermore, both the item content validity index (I-CVI) and the scale content validity index (S-CVI) for the total scale score and its four domains surpassed the threshold of 0.80. This signifies that all scales utilized in the study demonstrated acceptable levels of content validity.

## Discussion

This study provides valuable insights into diabetes prevention research by introducing a novel approach that empowers patients to serve as health educators for their family members. To our knowledge, this study represents the initial endeavor to investigate the empowerment of patients as health educators. The study's aims were to assess patients' willingness to take on this role, measure their confidence in their ability to succeed as health educators and examine the relationship between various factors in the Health Belief Model (HBM) and the success of this approach.

**Table 3. Descriptive statistics for variables, content validity and reliability for the scale.**

| Scale | M | SD | Min | Max | Cronbach's α | Inter-rater reliability coefficient (%) | I-CVI | S-CVI/UA |
|---|---|---|---|---|---|---|---|---|
| Total scores (14 questions) | 58.44 | 7.35 | 38 | 70 | .75 | 85 | 1.00 | 1.00 |
| Perceived Severity and Susceptibility. (2 questions) | 8.19 | 1.36 | 4 | 10 | .79 | 90 | 1.00 | 1.00 |
| Perceived Benefits and Perceived Barriers. (6 questions) | 24.66 | 3.47 | 16 | 30 | .79 | 87 | 1.00 | 1.00 |
| Cues to Action. (1 question) | 4.45 | .700 | 2 | 5 | 1.00 | 100 | 1.00 | 1.00 |
| Self-Efficacy. (5 questions) | 21.12 | 2.76 | 12 | 25 | .77 | 87 | 1.00 | 1.00 |

The study's findings reveal significant insights into the willingness and confidence of participants to take on the role of health educators for their family members and close relatives with regard to diabetes. A striking 87% of the participants expressed their willingness to serve as health educators for their family members and close relatives. This high percentage underscores a notable openness among patients to actively engage in health education within their immediate social circles. This willingness suggests a potential for leveraging patients as valuable resources in disseminating crucial information about diabetes prevention.

Another noteworthy finding is that the participants reported having a high self-efficacy level related to educating their family members and close relatives about diabetes. This level of self-efficacy among patients suggests that many of them perceive themselves as effective communicators and educators regarding diabetes-related information. This is particularly promising because confidence in one's ability to convey health-related messages is a critical factor in the success of health education efforts [4,16]. A notable correlation emerged between higher education levels and self-efficacy. This result aligns with existing literature, supporting the idea that education significantly contributes to enhancing self-efficacy [17].

Moreover, the study's findings shed light on the factors influencing participants' willingness to take on the role of health educators for their family members and close relatives regarding diabetes. These correlations provide valuable insights into the motivations and determinants that influence their readiness to engage in this novel approach.

Looking at the total score, the results highlight that significant differences were found in the total score of the tool based on the patient's education and income level. This aligns with existing literature suggesting a strong correlation between higher education levels and enhanced cognitive abilities or self-efficacy in various domains [18]. The disparity in scores emphasizes the potential influence of educational attainment on individuals' abilities to engage effectively with health-related information or tasks. Furthermore, these findings highlight the need for tailored interventions or educational approaches targeted at individuals with lower educational backgrounds. With regards to income status, this finding aligns with existing socio-economic research indicating that lower income levels often correlate with reduced access to resources, which can influence health-related knowledge or practices [19,20]. Individuals with limited financial means might face barriers to accessing quality healthcare or health-related information, potentially impacting their overall understanding or engagement with health-related measures. Moreover, this finding emphasizes the necessity for targeted interventions or support programs that consider the socio-economic context of individuals. Efforts to bridge this gap might involve initiatives offering accessible health-related information or resources tailored to individuals from lower-income brackets. Additionally, advocating for policies that address socio-economic inequalities could contribute to more equitable health outcomes across diverse income groups.

The analysis revealed significant associations between the perceived susceptibility and severity domain and several participant characteristics, namely their role within the family, employment status, education level, and family income. These disparities shed light on the diverse perceptions of susceptibility and severity regarding health concerns among different demographic groups.

The findings indicated a notable connection between participants' familial roles and their perceptions of susceptibility and severity. Grandmothers exhibited significantly lower mean scores in perceived susceptibility and severity compared to other family roles, such as sons, daughters, mothers, fathers, and grandfathers. This variation might stem from differing responsibilities or perspectives within the family structure, influencing their awareness or understanding of health-related risks. Moreover, employment status emerged as another influential factor. Unemployed participants displayed significantly lower mean scores in perceived

susceptibility and severity compared to full-time employees, part-time employees, retirees, and students. This finding echoes the literature that employment status as a significant role in the perceived susceptibility and severity [21–23]. This suggests that the nature of one's occupation or lack thereof might influence their perceptions of health risks and the severity of potential health outcomes. Similarly, education level played a crucial role, with participants holding a high school certificate reporting significantly lower mean scores in perceived susceptibility/severity compared to those with other education levels. This aligns with existing literature indicating a correlation between higher education and greater health literacy, potentially impacting individuals' perceptions of health risks [24]. Furthermore, family income disparities were associated with differences in perceived susceptibility/severity. This suggests that varying financial situations might influence individuals' perceptions of the severity of health-related issues, possibly due to differential access to healthcare or resources. These findings underscore the multifaceted nature of perceptions regarding health risks within diverse demographic groups. Tailored health interventions and educational programs considering these demographic variations might effectively address and mitigate disparities in health perceptions and enhance health-related understanding across different populations.

The analysis regarding the perceived benefits/barriers domain revealed significant associations with three key demographic factors: marital status, education level, and family income. These findings highlight the influence of these factors on individuals' perceptions regarding the advantages and obstacles associated with health behaviors or practices.

Marital status emerged as a significant factor influencing perceived benefits and barriers, illustrating notable differences among widowed participants in comparison to other individuals. Widowed individuals reported significantly lower mean scores in perceived benefits and barriers, indicating potentially unique perspectives on health-related advantages and challenges compared to other marital statuses. These findings align with existing literature. For instance, research suggests higher mortality rates among individuals who are non-married or previously married compared to those who are married [25]. This discrepancy might reflect a lower engagement in protective or healthy behaviors among non-married individuals, contributing to adverse health outcomes [25]. Also this was supported in the literature that marital status is an important predictor of healthcare utilization [26].

Education level exhibited notable disparities in perceived benefits and barriers. Participants with high diploma certificates reported significantly lower mean scores compared to those with uneducated/other levels of education. This suggests that varying educational backgrounds may significantly influence individuals' perceptions of the advantages and hindrances associated with health behaviors, potentially due to differences in health literacy or exposure to health-related information. Additionally, family income disparities were linked to variations in perceived benefits and barriers. Participants with mid-level family incomes reported significantly lower mean scores compared to those with lower or higher incomes. This findings is supported in the literature and underscores how differing financial circumstances might shape individuals' perceptions of the benefits and barriers related to health practices, likely influenced by disparities in access to healthcare resources or information [27–29].

These findings emphasize the intricate relationship between demographic factors and perceptions of the benefits and barriers associated with health behaviors. Understanding these variations is vital for tailoring targeted health interventions and educational programs that account for diverse perspectives stemming from different demographic backgrounds. Addressing these discrepancies can aid in the development of more effective strategies to promote health behaviors and overcome barriers across various population groups.

One significant finding is a positive and moderate correlation between participants' willingness to act as health educators and their perceptions of both the severity and susceptibility of

diabetes in their family members. This suggests that individuals who view diabetes as a significant health threat and consider their family members susceptible to it are more likely to volunteer as health educators. This underscores the pivotal role of individuals' perceptions of the severity of the health issue in motivating proactive health education, consistent with previous research [30–32].

A similar positive and moderate correlation emerged between individuals' willingness to serve as health educators and their perceptions of benefits and barriers. Those who perceive greater advantages in health education and encounter fewer obstacles tend to be more inclined to express their readiness to educate their family members and close relatives. Consequently, interventions that emphasize the advantages of health education and mitigate barriers have the potential to boost individuals' willingness to participate in such activities, as previous studies have demonstrated [33].

The study also revealed a strong and positive correlation between the willingness of individuals to become health educators and the presence of cues to action. Participants who reported receiving cues or prompts to engage in health education were more likely to express their readiness to educate their family members and close relatives. This underscores the significance of external triggers, reminders, or prompts in motivating individuals to embrace the role of health educators. These findings align with prior research that highlights the importance of triggers in maintaining self-management and promoting other favorable healthcare activities [34].

Finally, perhaps one of the most significant findings is the strong positive correlation between participants' willingness to be health educators and their self-efficacy. This suggests that individuals who have higher levels of self-efficacy and confidence in their ability to educate others are more inclined to embrace the role of health educators within their families. As participants' self-efficacy and confidence increased, their willingness to educate their family members and close relatives also increased.

These findings underscore the multifaceted nature of individuals' willingness to serve as health educators for their family members and close relatives. They highlight the importance of addressing perceptions, motivations, and self-belief when designing interventions aimed at promoting health education within communities. Moreover, they emphasize the need for healthcare providers and public health initiatives to foster self-efficacy and provide cues to action as strategies to engage patients effectively in health education efforts, particularly in the context of diabetes prevention. These findings collectively highlight the potential for a patient-centric approach in diabetes prevention efforts. Patients' willingness and confidence to take on the role of health educators for their family members and close relatives suggest an opportunity to empower individuals with diabetes to actively contribute to the health and well-being of their loved ones [16]. This approach not only capitalizes on the unique position of patients but also fosters a sense of shared responsibility within families and communities, which can be instrumental in combating the rising prevalence of diabetes.

Finally, these findings highlight the need for healthcare providers and public health initiatives to recognize and support patients in their roles as health educators. By providing the necessary resources, training, and guidance, healthcare systems can harness the potential of patients to extend the reach of diabetes prevention efforts and ultimately reduce the burden of this chronic condition within families and communities [16].

## Limitations

This study has several notable limitations that should be acknowledged. Firstly, it relies on a convenience sample rather than employing a random sampling technique, which makes it challenging to generalize the results to broader populations. Additionally, there is a potential

for a social desirability bias, as participants may have responded in ways they deemed favorable to the research team or societal expectations. Furthermore, the study tools used were not validated in a large sample, which may impact the reliability and validity of the measurements. Lastly, it is possible that highly motivated patients were more likely to respond to the questionnaire, while those with little interest in the topic may have been less inclined to participate, potentially introducing bias into the study. Nonetheless, it's important to note that the data collected successfully addresses all the study's objectives.

## Conclusion

This study has illuminated significant insights into empowering patients as health educators for their family members in the context of diabetes prevention. Our investigation aimed to assess patients' willingness, confidence, and the influencing factors related to this role, uncovering compelling findings. Notably, an overwhelming 87% of participants expressed willingness to act as health educators for their family members, indicating a high openness among patients to actively engage in health education. Moreover, participants exhibited high self-efficacy, feeling confident in their ability to effectively educate their relatives about diabetes. The study identified various factors influencing patients' willingness to serve as health educators, such as perceptions of severity, susceptibility, benefits, and barriers related to diabetes within their families. Additionally, cues to action and increased self-efficacy emerged as strong motivators for patients to take on this role. These findings underscore the multifaceted nature of individuals' readiness to act as health educators and stress the importance of addressing perceptions, motivations, and self-belief in designing effective health education interventions. Leveraging patients as educators not only capitalizes on their unique position but also fosters shared responsibility within families and communities, potentially instrumental in combating diabetes.

To optimize the impact of patients with diabetes as health educators, specific strategies are crucial. Develop comprehensive training modules to equip patients with effective communication skills for diabetes prevention and management. Practical examples and interactive sessions can enhance their teaching abilities. Moreover, create interactive online forums and support networks where patient educators can exchange experiences, successful approaches, and valuable insights. Platforms like virtual seminars or dedicated websites can amplify their collective impact. In addition, foster partnerships between healthcare institutions, community organizations, and patient advocacy groups is also important. These collaborations can broaden the reach of patient educators, leveraging varied platforms and resources for wider dissemination. Lastly, acknowledgment and incentives can play a crucial role, for example, recognize and celebrate the contributions of patient educators through public acknowledgments, awards, or incentives. Acknowledgment ceremonies or rewards for outstanding contributions can motivate others to embrace this crucial role. By incorporating these tailored strategies, the potential of patients with diabetes as health educators can be unlocked, significantly amplifying their positive influence on diabetes awareness, prevention, and care within communities.

## Supporting information

**S1 Data set.**
(SAV)

**S1 File.**
(PDF)

**S2 File.**
(PDF)

**S3 File.**
(DOC)

## Author Contributions

**Conceptualization:** Mona Alanazi, Eman Bajmal, Abeer Aseeri, Ghaida Alsulami.

**Data curation:** Mona Alanazi, Eman Bajmal, Abeer Aseeri.

**Formal analysis:** Mona Alanazi.

**Investigation:** Mona Alanazi, Ghaida Alsulami.

**Methodology:** Mona Alanazi, Eman Bajmal.

**Supervision:** Mona Alanazi.

**Validation:** Mona Alanazi.

**Visualization:** Mona Alanazi.

**Writing – original draft:** Mona Alanazi, Eman Bajmal, Abeer Aseeri, Ghaida Alsulami.

**Writing – review & editing:** Mona Alanazi, Eman Bajmal, Abeer Aseeri, Ghaida Alsulami.

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
