## [Decision Letter · Decision Letter 0]

6 Dec 2023

PONE-D-23-32756Empowering Patients with Diabetes for Health Educators' Role within Their Family Members: A Cross-sectional StudyPLOS ONE

Dear Dr. Alanazi,

Thank you for submitting your manuscript to PLOS ONE. After careful consideration, we feel that it has merit but does not fully meet PLOS ONE’s publication criteria as it currently stands. Therefore, we invite you to submit a revised version of the manuscript that addresses the points raised during the review process.

We look forward to receiving your revised manuscript.

Kind regards,

Maher Abdelraheim Titi

Academic Editor

PLOS ONE

Journal Requirements:

Additional Editor Comments :

Thank you for the submitting this manuscript to PLOS ONE. This study has a significant impact on diabetes prevention measures. However, I have several important concerns

Page 11-line 72-73: The author hypothesized that diabetes prevention programs would reduce healthcare system costs; however, this study did not examine the cost effectiveness of involving patients in the education program, consider rewrite it please.

This study was carried out on diabetic adults. This should be reflected in the title and the abstract (page 2, line 39).Were there Saudis and non-Saudis among the participants? Clarifying this point will be useful for future research.Pilot testing was conducted among 10% of the total required sample size. Were those patients included in the analysis?Page 13, line 266: the Cronbach's alpha coefficient of Perceived Benefits and Perceived Barriers was 0.77; however, table 3 reported a different value (0.79).Page 13, line 268: The Cronbach's alpha coefficient of the self-efficacy scale was 0.78, but table three reported a different value. Please correct.Page 12, line 240: The author reported the difference in self-efficacy, but the data was not presented or provided (one-way analysis of variance (ANOVA) and Tukey’s HSD post-hoc tests).Page 8, line 148: What is the cute off point that is considered a lower score? (i.e., indicates the need for interventions).Result: Table 1, the many significant findings were reported in table 3, the authors did not investigate and explained such as:Perceived benefits/barriers vs Marital Status /family incomePerceived benefits/barriers vs family incomeCues to Action (worries level) vs Education level/age.Perceived susceptibility vs Employment Status and education level.The author report “Participant’s willingness to be a health educator” as separate binary question, however, this question displayed in the survey tool as 5-point Likert scale.??“Vancouver” style should be used as per PLOS On submission guideline.

Reviewers' comments:

Reviewer's Responses to Questions

**Comments to the Author**

1. Is the manuscript technically sound, and do the data support the conclusions?

Reviewer #1: Yes

Reviewer #2: Yes

2. Has the statistical analysis been performed appropriately and rigorously? 

Reviewer #1: Yes

Reviewer #2: Yes

3. Have the authors made all data underlying the findings in their manuscript fully available?

Reviewer #1: Yes

Reviewer #2: Yes

4. Is the manuscript presented in an intelligible fashion and written in standard English?

Reviewer #1: Yes

Reviewer #2: Yes

5. Review Comments to the Author

Reviewer #1: I would alter language- instead of diabetic patient, please use people living with diabetes(always) especially as you are promoting a person centered approach

The subject is very topical and serves as an important opportunity to engage service users as educators. It is more likely that advice would be adhered to if delivered by peers and more so if family members.

I am glad it was pointed out that structured education prior to enlisting candidates was important.

Testing their knowledge and understanding would need to be considered in the next iteration.

Results of this approach such as hba1c etc would be valuable to assess the effectiveness of this approach

Reviewer #2: Thank you for providing opportunity to review the manuscript, “Empowering Patients with Diabetes for Health Educators' Role within Their Family Members: A Cross-sectional Study”, which may have potential impact for the diabetes self-management and care. However, there are some suggestions from my side to make it better.

Abstract

Line 42.

“Various statistical analyses …….”

If authors revise this sentence and be specific that would be great.

Introduction

Page 4, line 72 to 79.

This paragraph is unclear, initially, authors described the hypothesis regarding the benefits i.e., costs reduction. Then, feasibility etc., which is not connected to the previous paragraph too. This is because in the last sentence of the previous paragraph, you wrote about sense of empowerment, solidarity and shared responsibility.

Therefore, please revised this whole paragraph. Describe how it empower…? You can add cost reduction as an example “For example, previous study by …. Showed that cost was reduced by …. And don’t forget to give the references.

And finally make a connection with your study site… Why is this needed?

Page 9, line 171, Replace “will be asked” by “asked” and more grammatical correction is needed throughout the manuscript.

Page 11, line 210.

Could author describe in details of using all statistical analyses? For example, why ANOVA, for which variables?

Discussion

Discussion section required major revision. Authors should compare and contrast the findings with the previous studies conducted, which is missing in each and every finding.

Such as in page 14, line 284, your findings says that 87% participants were willing to server as a health educator. Are there any previous studies conducted related to these findings? Are the findings similar to your study? OR is it difference? Give reason… what will these findings indicate? Please reorganise your writing in this way.

Please follow same style for other major findings such as self-efficacy…

In result section, you mentioned significant difference in self-efficacy among the patient’s education and income level. You need to discuss these findings here, why is this difference in you study site? Is there any study to support this?

Conclusions

Can you conclude based on you study aim and put these all recommendations at the end.

Are patients willing to serve as a health educator? What are the influencing factors?

Thank you!!!

6. PLOS authors have the option to publish the peer review history of their article (what does this mean?). If published, this will include your full peer review and any attached files.

Reviewer #1: **Yes: **Naresh Kanumilli

Reviewer #2: No

---

## [Author Response · Author response to Decision Letter 0]

31 Jan 2024

Dear Reviewers and /Editors, 

Thank you for your feedback and thoughtful comments. We have carefully reviewed and addressed each of the points raised in your review. All requested revisions and modifications have been implemented as per your suggestions. Please find our detailed responses and the corresponding changes in the attached document.

We sincerely appreciate your time and valuable input in enhancing the quality of our work.

Best regards,

Dr. Mona Alanazi

---

## [Decision Letter · Decision Letter 1]

19 Feb 2024

Empowering Patients with Diabetes for Health Educators' Role within Their Family Members: A Cross-sectional Study

PONE-D-23-32756R1

Dear Dr. Alanazi,

We’re pleased to inform you that your manuscript has been judged scientifically suitable for publication and will be formally accepted for publication once it meets all outstanding technical requirements.

Kind regards,

Maher Abdelraheim Titi

Academic Editor

PLOS ONE

Additional Editor Comments (optional):

Reviewers' comments:

Reviewer's Responses to Questions

**Comments to the Author**

1. If the authors have adequately addressed your comments raised in a previous round of review and you feel that this manuscript is now acceptable for publication, you may indicate that here to bypass the “Comments to the Author” section, enter your conflict of interest statement in the “Confidential to Editor” section, and submit your "Accept" recommendation.

Reviewer #2: All comments have been addressed

2. Is the manuscript technically sound, and do the data support the conclusions?

Reviewer #2: Yes

3. Has the statistical analysis been performed appropriately and rigorously? 

Reviewer #2: Yes

4. Have the authors made all data underlying the findings in their manuscript fully available?

Reviewer #2: Yes

5. Is the manuscript presented in an intelligible fashion and written in standard English?

Reviewer #2: Yes

6. Review Comments to the Author

Reviewer #2: Dear Authors, Thank you for sharing the revised version of the manuscript. This reads well for me. Best wishes from my side.

7. PLOS authors have the option to publish the peer review history of their article (what does this mean?). If published, this will include your full peer review and any attached files.

Reviewer #2: No

---

## [Editor Report · Acceptance letter]

4 Apr 2024

PONE-D-23-32756R1 

PLOS ONE

Dear Dr. Alanazi, 

I'm pleased to inform you that your manuscript has been deemed suitable for publication in PLOS ONE. Congratulations! Your manuscript is now being handed over to our production team.

Kind regards, 

on behalf of

Dr. Maher Abdelraheim Titi 

Academic Editor

PLOS ONE